# NMR-Based Metabolomic Analysis for the Effects of Trimethylamine N-Oxide Treatment on C2C12 Myoblasts under Oxidative Stress

**DOI:** 10.3390/biom12091288

**Published:** 2022-09-13

**Authors:** Hong Zou, Caihua Huang, Lin Zhou, Ruohan Lu, Yimin Zhang, Donghai Lin

**Affiliations:** 1School of Sport Science, Beijing Sport University, Beijing 100084, China; 2Physical Education Department, Xiamen University, Xiamen 361005, China; 3Research and Communication Center of Exercise and Health, Xiamen University of Technology, Xiamen 361021, China; 4Key Laboratory for Chemical Biology of Fujian Province, MOE Key Laboratory of Spectrochemical Analysis & Instrumentation, College of Chemistry and Chemical Engineering, Xiamen University, Xiamen 361005, China; 5Key Laboratory of Prevention and Treatment of Cardiovascular and Cerebrovascular Diseases, Ministry of Education and Key Laboratory of Biomaterials and Biofabrication in Tissue Engineering of Jiangxi Province, Gannan Medical University, Ganzhou 341000, China; 6Key Laboratory of Ministry of Education of Exercise and Physical Fitness, Beijing Sport University, Beijing 100084, China

**Keywords:** TMAO, C2C12 myoblasts, NMR-based metabolomics, oxidative stress

## Abstract

The gut microbial metabolite trimethylamine N-oxide (TMAO) has received increased attention due to its close relationship with cardiovascular disease and type 2 diabetes. In previous studies, TMAO has shown both harmful and beneficial effects on various tissues, but the underlying molecular mechanisms remain to be clarified. Here, we explored the effects of TMAO treatment on H_2_O_2_-impaired C2C12 myoblasts, analyzed metabolic changes and identified significantly altered metabolic pathways through nuclear magnetic resonance-based (NMR-based) metabolomic profiling. The results exhibit that TMAO treatment partly alleviated the H_2_O_2_-induced oxidative stress damage of cells and protected C2C12 myoblasts by improving cell viability, increasing cellular total superoxide dismutase capacity, improving the protein expression of catalase, and reducing the level of malondialdehyde. We further showed that H_2_O_2_ treatment decreased levels of branched-chain amino acids (isoleucine, leucine and valine) and several amino acids including alanine, glycine, threonine, phenylalanine and histidine, and increased the level of phosphocholine related to cell membrane structure, while the TMAO treatment partially reversed the changing trends of these metabolite levels by improving the integrity of the cell membranes. This study indicates that the TMAO treatment may be a promising strategy to alleviate oxidative stress damage in skeletal muscle.

## 1. Introduction

Skeletal muscle is one of the most dynamic and plastic tissues of the human body. It accounts for 40% of the total body weight and 50–70% of all body proteins, and significantly contributes to multiple bodily functions, such as movement, posture maintenance, heat generation, and breathing [1]. Mitochondria in skeletal muscle can produce reactive oxygen species (ROS) during normal metabolic processes [2]. Under normal circumstances, the generation and clearance of ROS in the body are in dynamic equilibrium. When the production rate of ROS is greater than the scavenging capacity, the body experiences oxidative stress. Excessive oxidative stress can cause impairments of myofiber functions. Previous studies have shown that during exercise, especially long-term and intensive exercise, endogenous reactive oxygen free radicals are rapidly increased and substantially accumulated in the body. The accumulated free radicals could interact with macromolecules, such as phospholipids, nucleic acids and side chains of polyunsaturated fatty acids which are associated with enzymes and membrane receptors, increasing lipid peroxidation products including malondialdehyde (MDA) and 4-hydroxynonenoic acid. The increased lipid peroxidation products might change the fluidity and permeability of the cell membrane and affect the normal structure and function of cells [3]. Therefore, it is quite necessary to develop efficient strategies to reduce oxidative stress in skeletal muscle.

Trimethylamine N-oxide (TMAO), a small organic molecule with a molecular weight of 75.11 Da, exists in animals, plants and fungi [4]. Notably, marine animals show higher levels of TMAO than other organisms [5]. As it is known, TMAO in the human body mainly comes from food being rich in substances with chemical structures of choline or carnitine or trimethylamine, such as betaine, phosphatidylcholine (PPC) and L-carnitine existing in seafood, fish, red meat and eggs. Once it enters the intestine, these substances are primarily metabolized into trimethylamine (TMA) [6,7], a small amount of dimethylamine (DMA) and methylamine (MA) [8,9] under the action of intestinal microflora. If TMA is not fully absorbed through the intestinal circulation, it will be further oxidized to TMAO under the action of flavin-containing monooxygenase 3 (FMO3) in the liver [6,7,10]. Recently, TMAO has attracted increasing attention due to its close relationship with cardiovascular disease [11] and type 2 diabetes [12,13]. The elevated TMAO level in sera has been identified as an indicator of metabolic syndrome [14], which might be related to a compensatory mechanism in response to the disease [15]. Furthermore, a series of studies have demonstrated that TMAO treatment reduces the negative effects of oxidative stress in the endoplasmic reticulum of streptozotocin diabetic rats [16] and MC65 human neuroblastoma cells [17]. However, it is still unclear whether TMAO treatment could reduce the negative effects of oxidative stress in skeletal muscle. If it could, what are the underlying metabolic mechanisms?

Metabolomic analysis has been extensively applied to exploit metabolic mechanisms underlying biological processes [18,19] and the beneficial effects of nutritional supplements [20]. Acting as a powerful detection technique, NMR spectroscopy plays an important role in metabolomic analysis with several advantages including high reproducibility, objective and quantitative measurements, simple and nondestructive sample pretreatment, as well as fast and easy sample detection [21].

Belonging to myogenic cells from skeletal muscle satellite cells, C2C12 myoblasts are widely used to establish murine muscle cell models to explore muscle regeneration. Here, we established a H_2_O_2_-induced oxidative stress model of cells [22,23] and performed NMR-based metabolomic analyses to address the effects of both H_2_O_2_ treatment and TMAO treatment on C2C12 cells. We found that TMAO treatment could partially reverse the changing trends of some metabolite levels impaired by H_2_O_2_ treatment. Our results shed light on the molecular mechanisms underlying the beneficial effects of TMAO treatment for alleviating oxidative stress in skeletal muscle.

## 2. Materials and Methods

### 2.1. Reagents

H_2_O_2_ was purchased from Sinopharm Chemical Reagent Co. Ltd, Shanghai, China (30%, 10011218). TMAO was purchased from the Sigma-Aldrich, St. Louis, MI, USA (95%, 317594-5G), T-SOD assay kit (A001-1-2) and MDA assay kit (A003-1-2) were purchased from Nanjing Jiancheng Bioengineering Institute (Nanjing, China). Details of the used antibodies are shown as follows: MyoD1 (ab64159, Abcam, Cambridge, UK), CAT (21260-1-AP, Proteintech, Wuhan, China) and GAPDH (10494-1-AP, Proteintech, Wuhan, China).

### 2.2. Cell Culture and Experimental Design

Murine skeletal muscle-derived C2C12 myoblasts were purchased from the Stem Cell Bank, Chinese Academy of Sciences, China (Shanghai, China). Cells were grown in standard culture media containing Dulbecco’s modified Eagle’s medium (DMEM; HyClone, Logan, UT, USA) combined with 10% fetal bovine serum (Gibco, Gaithersburg, MD, USA), 100 U/mL penicillin and 100 mg/mL streptomycin. Cells were cultured in a humidified atmosphere of 5% CO_2_ at 37 °C.

The cell experimental design is displayed in Appendix A. C2C12 cells were first cultured in medium to 50–60% confluency, H_2_O_2_ was added to the cells at a final concentration of 0.5 mM, and the cells were further incubated for 2 h. Then, the medium was removed from the cells, TMAO dissolved in fresh medium was added into the cells at a final concentration of 1 mM, 5 mM or 10 mM, and thereafter the cells were incubated for 24 h.

The C2C12 cells were divided into five groups: normal (Nor) group, H_2_O_2_-treated (H_2_O_2_) group, 1 mM TMAO-treated (TMAO1) group, 5 mM TMAO-treated (TMAO5) group, and 10 mM TMAO-treated (TMAO10) group. These five groups were used for subsequent assays and WB analysis. The TMAO5 group displayed the best beneficial effects on H_2_O_2_-impaired C2C12 cells. Thus, the three following groups of C2C12 cells were used for further metabolomic analysis: Nor, H_2_O_2_ and TMAO5.

### 2.3. MTS Cell Viability Assay and Cell Proliferation Determination

To measure the cell viability, C2C12 cells were seeded in 96-well plates and cultured as previously described [24]. Then, 20 μL of the solution used in the CellTiter 96 AQueous One Solution Cell Proliferation Assay Kit (MTS, Promega, Madison, WI, USA) was added to each well. The cells were incubated in the dark at 37 °C for 3 h before measuring the absorbance of formazan at 490 nm on a microplate reader (BioTek, Winooski, VT, USA). Finally, the C2C12 cells were randomly selected and scanned on a fluorescence microscope (Motic, Xiamen, China).

We analyzed the proliferation abilities of the cells incubated for 24 h using the cell counting method. The adherent C2C12 cells were washed three times with phosphate-buffered saline (PBS) to remove dead cells, 1 mL of 0.25% trypsin was added for a 1 min digestion, and 1 mL of fresh DMEM was added. The cells were carefully suspended, placed into 2 mL Eppendorf tubes, and centrifuged at 1000 rpm for 30 s in a microcentrifuge (Mini-4K, CENCE, CHN). Then, the culture medium was removed and the cells were resuspended in 1 mL of PBS. After being diluted 10 times, 20 μL of the cell suspensions were transferred into a counting plate (CO010101, Countstar, Shanghai, China), and the number of cells was determined with an automatic cell counter (IC1000, Countstar).

### 2.4. Measurement of Cellular T-SOD Capacity

The C2C12 cells were collected in centrifuge tubes according to the protocol described in the T-SOD assay kit instruction. Reagents were added to the tubes following the manufacturer’s instructions, and the tubes were then placed at room temperature for 10 min. Absorbance values of cell samples were detected at a 1 cm optical path by a multi-mode microplate readers (POLAR 4star Omega, Offenburg, Germany) at 550 nm, and the total SOD activity of each cell sample was calculated based on its absorbance value.

### 2.5. Measurement of Cellular MDA Level

The C2C12 cells were collected following the protocol described in the MDA assay kit instructions. Both the cells and test reagents were added to the centrifuge tubes and mixed with a vortex mixer. The samples were placed in a water bath at 95 °C for 40 min, and thereafter, cooled with running water and centrifuged at 3500 rpm for 10 min, and then the supernatant was removed. The absorbance value of each tube was detected at a 1 cm optical path by a multi-mode microplate reader (POLAR 4star Omega, Offenburg, Germany) at 532 nm.

### 2.6. Western blotting

The C2C12 cell lysates were collected using RIPA lysis buffer with a protease inhibitor and a phosphorylation protease inhibitor cocktail (Thermo Fisher, Waltham, MA, USA), sonicated for 35 s, and centrifuged (11,000× *g*, 10 min, 4 °C). Then, the cell supernatants were collected and protein concentrations were determined using a BCA Protein Analysis Kit (Beyotime, Shanghai, China). The cell lysates were resolved using sodium dodecyl sulfate–polyacrylamide (SDS-PAGE) and transferred onto PVDF membranes (GE, Freiburg, Germany). After being blocked with 5% non-fat powdered milk for 1 h, the PVDF membranes were probed with antibodies including MyoD1, CAT and GAPDH. Using horseradish peroxidase-conjugated antibodies, the proteins were visualized by enhanced chemiluminescence. Subsequently, the band densities were analyzed with the ImageJ software (National Institutes of Health, Bethesda, Rockville, MD, USA).

### 2.7. Extraction of Intracellular Aqueous Metabolites

Following the protocol described in our previous work [20], aqueous metabolites were extracted from C2C12 myoblasts for NMR-based metabolomic analysis. Approximately 6 × 10^6^ cells were harvested from 10 cm diameter culture dishes. After the medium was removed, the cells were rapidly rinsed with pre-cooled 4 °C PBS (pH 7.4) three times to reduce the residual medium. Then, methanol was immediately added to quench the cells, and the cells were scraped with a cell scraper into a 15 mL centrifuge tube. Chloroform and ultrapure water were added according to a volume ratio of methanol, chloroform, and water of 4:4:2.85. To evenly mix the samples, the 15 mL centrifuge tube was vortexed for 5 min, placed on ice for 30 min, and then centrifuged (12,000× *g*, 15 min, 4 °C). Subsequently, the aqueous solution in the upper layer was extracted by blowing dry the methanol with a nitrogen blower, and finally lyophilized into solid powder with a freeze dryer.

### 2.8. Sample Preparation and NMR Measurements

The solid powder was suspended in 550 µL of NMR buffer (50 mM, pH 7.4, 100% D_2_O, and 0.05 mM sodium 3-(trimethylsilyl) propionate-2,2,3,3-d4 (TSP)), vortexed, and then centrifuged at 12,000× *g* for 15 min at 4 °C. Then, all samples were transferred into 5 mm NMR tubes for the following NMR metabolomic experiments.

All NMR measurements were performed at 298 K on a Bruker Avance III 850 MHz NMR spectrometer (Bruker Bio Spin, Rheinstetten, Germany) equipped with a TCI cryoprobe. One dimensional (1D) ^1^H spectra were acquired using the pulse sequence NOESYGPPR1D [RD-G1-90°-t1-90°-τm-G2-90°-ACQ] with water suppression, t1 (short delay) was 4 μs, and τ_m_ (mixing time) was 10 ms. Pulsed gradients G1 and G2 were used to improve the quality of water suppression. A total of 128 transients were collected into 64 k data points using a spectral width of 20 ppm with an acquisition time (ACQ) of 1.93 s. To confirm the resonance assignments of the metabolites, two-dimensional (2D) ^1^H-^13^C heteronuclear single quantum coherence (HSQC) spectrum and 2D ^1^H-^1^H total correlation spectroscopy (TOCSY) spectrum were recorded according to the referenced protocol [20]. The 2D ^1^H-^13^C HSQC spectrum was recorded with a spectral width of 10 ppm in the ^1^H dimension and 110 ppm in the ^13^C dimension, a data matrix of 1024 × 128 points, and a relaxation delay of 1.5 s. The 2D ^1^H-^1^H TOCSY spectrum was recorded with a spectral width of 10 ppm in both ^1^H dimensions, a data matrix of 2048 × 256 points and a relaxation delay of 1.5 s.

### 2.9. NMR Data Preprocessing

On-dimensional NMR spectra were processed for phase correction, baseline correction and resonance alignment with the MestReNova 9.0 software (Mestrelab Research S.L., Santiago de Compostela, Spain). The spectra region of δ 5.15–4.85 (water resonance) was then excluded to eliminate the influence of the water peak on the spectral line integration, and the region of δ 9.5–0.8 was segmented into bins with a width of 0.001 ppm for statistical analysis. Peak integrals for each NMR spectrum were normalized by the TSP spectral integral to represent the relative levels of assigned metabolites.

### 2.10. NMR Data Analysis

The NMR resonances of metabolites were assigned by using a combination of Chenomx NMR Suite, the HMDB, and previously published literature, which were confirmed by using 2D ^1^H-^13^C HSQC and ^1^H-^1^H TOCSY spectra. Then, multivariate statistical analyses were performed with the SIMCA-P software (version 14.1.0, MKS Umetrics, Umea AB, Sweden). Pareto scaling was applied to enhance the significance of low-level metabolites without amplifying noise. Unsupervised principal component analysis (PCA) was performed to verify the metabolic separation and show clusters among the samples. Moreover, supervised partial least-squares discriminant analysis (PLS-DA) was conducted to check grouping trends and improve the separation between metabolic profiles. Cross-validation was conducted to evaluate the robustness of the PLS-DA model (*n* = 200) with two model parameters: R2 and Q2. The R2 and Q2 values approaching 1 reflected an increasing reliability of the PLS-DA model.

Based on the metabolite integrals, we quantitatively compared the relative levels of the assigned metabolites between the groups by performing a two-tailed Student’s *t*-test with GraphPad Prism software (version 8.3.0, La Jolla, CA, USA). Data were represented as the mean ± standard deviation (SD). Differential metabolites were identified with the criterion of statistical significance (*p* < 0.05).

Furthermore, we conducted metabolic pathway analysis using the MetaboAnalyst 5.0 (https://www.metaboanalyst.ca, accessed on 7 March 2022) provided by McGill university in Montreal, Canada. We implemented the pathway topology analysis with a combination of metabolite set enrichment analysis and pathway topological analysis. Two criteria of *p* < 0.05 and pathway impact value (PIV) >0.2 were used to identify the significantly altered metabolic pathways. Additionally, we conducted hierarchical cluster analysis for confirming the metabolic separation of the three groups of C2C12 cells with MetaboAnalyst 5.0.

## 3. Results

This section may be divided by subheadings. It should provide a concise and precise description of the experimental results, their interpretation, as well as the experimental conclusions that can be drawn.

### 3.1. Establishment of the Oxidative Stress Model of C2C12 Cells

The oxidative stress model of C2C12 cells was established through H_2_O_2_-inducing cytotoxicity [22,23]. We separately treated C2C12 cells by H_2_O_2_ at 0.1, 0.25, 0.5, 1 and 2 mM for 2 h, and found that the H_2_O_2_ treatments at concentrations of ≥0.5 mM significantly decreased the viabilities of the normal cells (*p* < 0.001) (Appendix A). Thus, we establish the oxidative stress model of C2C12 cells by treating the cells by H_2_O_2_ at 0.5 mM for 2 h.

### 3.2. Effect of TMAO Treatment on the Proliferation Ability of C2C12 Cells

We examined the effects of TMAO treatment on the proliferation abilities of normal C2C12 cells in vitro using various concentrations of TMAO (0.1, 0.5, 1, 5, 10, 50 and 100 mM) and different treatment times (24, 48 and 72 h). The proliferation abilities of the cells treated by 0.1, 0.5, 1 and 5 mM TMAO for 24 h were not observably decreased relative to normal cells, while those treated by 10, 50 and 100 mM TMAO were significantly decreased (Figure 1A). Expectedly, extending the treatment time would decrease the TMAO concentration required to significantly inhibit the cell proliferation. Either the 48 h TMAO treatment at a concentration of ≥0.1 mM (Figure 1B), or the 72 h TMAO treatment at a concentration of ≥0.5 mM (Figure 1C) significantly decreased the cell proliferation ability. We further performed the following experiments on H_2_O_2_-impaired C2C12 cells undergoing the 24 h TMAO treatment at one of the three following concentrations of 1, 5 and 10 mM, which were defined as the TMAO1, TMAO5 and TMAO10 groups, respectively.

### 3.3. TMAO Treatment Partly Enhanced the Proliferation and Differentiation Potential of H_2_O_2_-Impaired C2C12 Cells

Significantly, the 24 h TMAO treatments at 1, 5 and 10 mM all increased the proliferation abilities of H_2_O_2_-impaired C2C12 cells, and 5 mM TMAO treatment exhibited the maximal beneficial effect (Figure 2F). To assess the dependence of the effect of TMAO treatment on the treatment time, we quantitatively compared the cell viabilities among normal cells, H_2_O_2_-impaired cells, and H_2_O_2_-impaired cells with 5 mM TMAO treatment for 4 h, 12 h and 24 h separately. We found that all the three TMAO treatments could partly recover the viabilities of the H_2_O_2_-impaired C2C12 cells, and the longer treatment time showed a more significant beneficial effect on the proliferation of H_2_O_2_-impaired C2C12 cells (Figure 2G).

The expression level of myogenic differentiation 1 (MyoD1) protein usually reflects the differentiation potential of myoblasts [20]. We quantitatively analyzed the expression levels of MyoD1 in Normal, H_2_O_2_, TMAO1, TMAO5 and TMAO10 cells. While the H_2_O_2_ group showed a significantly decreased expression level of MyoD1 compared to the Normal group, the TMAO1, TMAO5 and TMAO10 groups exhibited profoundly increased MyoD1 expression levels compared with the H_2_O_2_ group (Figure 2H).

### 3.4. TMAO Treatment Enhanced Antioxidant Activities in H_2_O_2_-Impaired C2C12 Cells

Both SOD and CAT are antioxidant enzymes capable of protecting cells from oxidative stress damage. We investigated the effect of TMAO on antioxidant activities in H_2_O_2_-impaired C2C12 cells by measuring the activity of T-SOD and the expression level of CAT. Compared with normal cells, the H_2_O_2_-impaired cells showed a significantly decreased activity of T-SOD, while the three TMAO treatments profoundly increased the activities of T-SOD in the cells (Figure 3A). The H_2_O_2_ group showed a remarkably decreased expression level of CAT compared to the normal group, but the three TMAO-treated groups displayed significantly increased expression levels of CAT compared to the H_2_O_2_ group (Figure 3B,C). Notably, the TMAO5 group showed the most significant increase in the CAT expression, and also the most significant effect on the proliferation ability of the C2C12 cells (Figure 2F).

### 3.5. TMAO Treatment Decreased the MDA Level of H_2_O_2_-Impaired C2C12 Cells

MDA is a lipid peroxidation product formed by the combination of accumulated reactive oxygen free radicals and macromolecular substances on the cell membrane that trigger lipid peroxidation under oxidative stress. This might affect the normal structure and functions of cells and cause oxidative stress damage [3]. We assessed the effect of TMAO treatment on MDA generation in H_2_O_2_-impaired C2C12 cells. While the H_2_O_2_ group showed an obviously increased level of intracellular MDA compared with the normal group, either 1 mM or 5 mM TMAO treatment significantly decreased the MDA level in H_2_O_2_-impaired C2C12 cells (Figure 3D). These results indicate that the TMAO treatment could alleviate the H_2_O_2_-induced oxidative stress damage in the cells by partly scavenging oxidative stress products.

### 3.6. NMR Spectra of Aqueous Extracts Derived from C2C12 Myoblasts

We performed the NMR-based metabolomic analysis to address the metabolic mechanisms of TMAO treatment for alleviating H_2_O_2_-induced oxidative stress damage in C2C12 cells. Figure 4 displays the typical 850 MHz ^1^H NMR spectra recorded on aqueous extracts derived from the normal, H_2_O_2_ and TMAO5 groups of C2C12 cells. A total of 45 metabolites were identified (Appendix A). The resonance assignments of the metabolites were confirmed by using 2D ^1^H-^13^C HSQC and ^1^H-^1^H TOCSY spectra (Appendix A). The NMR spectra illustrated that the TMAO treatment led to a remarkable accumulation of intracellular TMAO (Appendix A).

### 3.7. Multivariate Data Analysis for NMR Data of C2C12 Myoblasts

To explore the differences between the metabolic profiles of the three groups of C2C12 cells, we performed multivariate data analysis on NMR spectral data. The unsupervised PCA model with the first two principal components t[1] and t[2] was constructed to observe the trend of the overall metabolic profiles. The PCA scores plot shows distinct metabolic profiles of the three groups (Figure 5A). In addition, the metabolic profile of the H_2_O_2_ group was distinctly distinguished from the normal group along t1 (Figure 5B), and the metabolic profile of the TMAO5 group was distinctly distinguished from the H_2_O_2_ group along t[1] (Figure 5C). To further verify the reliability of the PCA model, we also carried out hierarchical cluster analysis using MetaboAnalyst 5. The hierarchical cluster analysis displayed that the normal, H_2_O_2_ and TMAO5 groups were divided into three different clusters (Figure 5D), similarly to the PCA scores plot, confirming that the PCA model was reliable.

Additionally, we conducted supervised PLS-DA analyses between the three groups of the C2C12 cells to improve the metabolic separations. The constructed PLS-DA scores plots display that the metabolic profile of the H_2_O_2_ group was clearly distinguished from those of the normal and TMAO5 groups. The cross-validation plots were indicative of the higher robustness and reliability of these two PLS-DA models (Figure 6). These results show that the H_2_O_2_ treatment profoundly impaired the metabolic profile of C2C12 cells, while the TMAO treatment partly restored the impaired metabolic profile of the cells.

### 3.8. Identifications of Differential Metabolites

To address the changes in metabolite levels between the three groups of C2C12 cells, we calculated the relative levels of the identified metabolites based on their relative NMR integrals. We compared the metabolite levels between H_2_O_2_ and normal groups, and TMAO5 and H_2_O_2_ groups using one-way ANOVA analysis followed by Tukey’s multiple comparison test (Table 1). For the H_2_O_2_ group vs. the normal group, 31 differential metabolites were identified, including 12 increased metabolites (acetate, glutamate, methionine, sarcosine, glutathione, taurine, myo-Inositol, creatine, phosphocholine (PC), glucose, UDP-GlcNAc and formate), and 19 decreased metabolites (pantothenate, isoleucine, leucine, valine, alanine, putrescine, TMA, glycine, threonine, lactate, carnosine, tyrosine, phenylalanine, histidine, GTP, AMP, NADPH, ADP and IMP). For the TMAO5 group vs. the H_2_O_2_ group, 30 differential metabolites were identified, including 16 increased metabolites (pantothenate, isoleucine, leucine, valine, alanine, putrescine, TMA, TMAO, glycine, lactate, glucose, tyrosine, phenylalanine, GTP, formate and IMP), and 14 decreased metabolites (ethanol, acetate, glutamate, glutamine, methionine, sarcosine, glutathione, lysine, choline, sn-Glycero-3-phosphocholine (GPC), myo-Inositol, creatine, PC and UDP-GlcNAc) (Table 1 and Figure 7).

### 3.9. Identification of Significantly Altered Metabolic Pathways

We performed metabolic pathway analyses based on the levels of the identified metabolites in the three groups of the C2C12 cells (Figure 8). From the pair-wise comparisons of the H_2_O_2_ group vs. the normal group, and the TMAO5 group vs. the H_2_O_2_ group, we identified the following significantly altered metabolic pathways, including (1) glutathione metabolism; (2) D-glutamine and D-glutamate metabolism; (3) alanine, aspartate and glutamate metabolism; (4) purine metabolism; (5) histidine metabolism; (6) arginine and proline metabolism; (7) glycine, serine and threonine metabolism; (8) phenylalanine, tyrosine and tryptophan biosynthesis; (9) phenylalanine metabolism; (10) beta-alanine metabolism; (11) taurine and hypotaurine metabolism; and (12) nicotinate and nicotinamide metabolism.

Compared with the normal group, the H_2_O_2_ group showed 10 significantly altered metabolic pathways (1, 3, 5, 6, 7, 8, 9, 10, 11 and 12). Furthermore, compared to the H_2_O_2_ group, the TMAO5 group displayed 6 of the 10 significantly altered metabolic pathways (1, 3, 5, 6, 7 and 9) with 2 extra metabolic pathways (2 and 4).

The Kyoto Encyclopedia of Genes and Genomes (KEGG) has been widely used as an important database to reconstruct metabolic networks and identify significant metabolic pathways. To systematically visualize remarkably changed metabolites in both H_2_O_2_-impaired cells and H_2_O_2_-impaired cells with TMAO treatment, we projected pivotal metabolites onto a metabolic map based on the KEGG database (Figure 9). Overall, the H_2_O_2_ group showed markedly impaired metabolite changes, and the TMAO5 group partly reversed these impaired metabolite changes.

## 4. Discussion

It is known that the proliferation of muscle satellite cells can provide a sufficient number of muscle progenitor cells, contributing to muscle growth, repair and adaptation to stress (including exercise, disease, injury and aging) [25]. As the foundation of muscle growth and regeneration, the normal functions of muscle satellite cells are crucial for maintaining the stability of the internal environment of skeletal muscle [26]. In this study, we exploited the effects of TMAO treatment on H_2_O_2_-impaired C2C12 myoblasts, and addressed the underlying metabolic mechanisms. H_2_O_2_ treatment greatly decreases the proliferation and differentiation potential of C2C12 cells, inducing oxidative stress damage. Significantly, the TMAO treatment partly recovers the decreased proliferation and differentiation potential of the H_2_O_2_-impaired C2C12 cells, and significantly enhances the antioxidant activities in the H_2_O_2_-impaired cells. Moreover, TMAO treatment alleviates the H_2_O_2_-induced oxidative stress damage in the cells by partly scavenging oxidative stress products. Furthermore, the TMAO treatment significantly altered 6 of 10 H_2_O_2_-impaired metabolic pathways including: glutathione metabolism; alanine, aspartate and glutamate metabolism; histidine metabolism; arginine and proline metabolism; glycine, serine and threonine metabolism; phenylalanine metabolism.

As one of myogenic regulatory factors, the MyoD1 protein has been previously identified to be a marker of terminal specification in the muscle lineage [27,28,29]. In this study, we found that the H_2_O_2_ treatment (0.5 mM) significantly decreased the proliferation ability of C2C12 cells and obviously down-regulated the expression level of MyoD1, while TMAO treatment partly reversed the proliferation and differentiation potential of the C2C12 cells and expression level of MyoD1 in the H_2_O_2_-impaired cells.

Furthermore, alkaloids such as carnitine, choline, and betaine can be transformed into TMA under catalysis by intestinal microbiota, which are either partially absorbed in the colon or transformed into TMAO by flavin monooxygenase (mainly FMO3) in the liver [6,7,8,9,10]. Furthermore, humans can also uptake TMAO through fishes and other sea products, which are absorbed in intestine via gut microflora [30,31,32]. A previous study of 40 healthy young men showed that fish consumption led to a peak plasma concentration of TMAO 2 h after the meal with a plasma half-life of 4 h, and approximately 19% of TMAO were detected in skeletal muscle in 6 h [31]. Additionally, we showed that the content of TMAO in skeletal muscle of mice with the ingestive gavage of TMAO peaked at 15 min, keeping a high level for 1 h. These results indicated that the TMAO can result from the transformation of alkaloids catalyzed by FMOs, and can also enter plasma and skeletal muscle rapidly. Nevertheless, the biological effects of TMAO still remain to be clarified.

As one of lipid peroxidation products, MDA, can aggravate the damage of cell membranes, the MDA concentration indirectly reflects the degree of cell damage. In this study, the H_2_O_2_ group displayed distinctly down-regulated levels of T-SOD and expression level of CAT, while the TMAO1, TMAO5 and TMAO10 groups exhibited obviously up-regulated levels of T-SOD and expression level of CAT (Figure 3). Moreover, H_2_O_2_ treatment dramatically increased the MDA level in C2C12 cells, which could be significantly reversed by 1 mM and 5 mM TMAO treatments. These results show that TMAO treatments at suitable concentrations could significantly alleviate the oxidative stress damage in H_2_O_2_-impaired C2C12 cells and profoundly enhance the antioxidant capacity of the impaired cells.

Previously, several animal studies reported neutral or even beneficial effects of TMAO or its precursors in cardiovascular disease model systems, supporting the clinically proven beneficial effects of either its precursor, L-carnitine, or a sea-food rich diet (naturally containing TMAO) on cardiometabolic health [15]. It was also reported that TMAO at the dose of 75 mM could inhibit the level of F2-isoprostane in the human neuroblastoma cell line MC65 to a great extent, and played a chemical chaperone function to promote protein function [17]. Additionally, a previous study reported the protective effect of TMAO treatment on the INS-1 β-Cell under diabetic glucolipotoxic conditions [33]. Thus, the beneficial role of TMAO at the cellular level was emphasized by the above studies. In particular, TMAO has been widely recognized as a chemical chaperone to promote protein folding and maintain protein structure and function.

Amino acids are the fundamental units of proteins, particularly the branched-chain amino acids (BCAAs) including leucine, valine and isoleucine. They cannot only stimulate protein synthesis in muscle, but also alleviate muscle atrophy [34,35]. In this study, we found that H_2_O_2_ treatment decreased the levels of BCAAs and other amino acids (alanine, glycine, threonine, phenylalanine and histidine) in C2C12 myoblasts, while the TMAO treatment could partly restore the levels of these amino acids (Table 1). Previous studies have revealed the important role of alanine metabolism in the human body, and identified alanine to be the main substrate for amino acid transformation and gluconeogenesis by the activation of muscle protein catabolism [36,37]. In this study, the TMAO5 group displayed an increased level of alanine compared to the H_2_O_2_ group, which was even higher than that of the normal group.

It was previously reported that the TMAO treatment could improve the deficiency of energy and amino acid transformation substrate in damaged cells via upregulating alanine, and tyrosine and phenylalanine in skeletal muscle are involved in protein synthesis and degradation [38]. In this study, the TMAO5 group displayed significantly increased levels of tyrosine and phenylalanine compared to the H_2_O_2_ group (*p* < 0.001), indicating that the TMAO treatment could reverse the suppressed protein synthesis in H_2_O_2_-impaired C2C12 cells. Furthermore, the TMAO treatment significantly altered 6 of the 10 H_2_O_2_-impaired metabolic pathways, including amino acid metabolism pathways: alanine, aspartate and glutamate metabolism; histidine metabolism; glycine, serine and threonine metabolism; phenylalanine, tyrosine and tryptophan biosynthesis; and phenylalanine metabolism. The results above supported the conclusion that TMAO is a protein stabilizer.

Additionally, our study also showed that H_2_O_2_ treatment up-regulated the MDA level in C2C12 myoblasts, leading to the instability or damage of cell membranes [39,40]. As well known, PPC is the most abundant phospholipid in cell membranes. The essential precursors of PPC are choline and PC, and the decomposition product of PPC is GPC [41]. Upon lipid peroxidation destroying lipid membranes, the promotion of PPC synthesis is a prerequisite for maintaining membrane integrity [24]. Compared with the normal group, the H_2_O_2_ group displayed the increased level of PC, and basically unchanged levels of choline and GPC. Significantly, the TMAO treatment reduced the level of lipid peroxidation, and dramatically decreased the levels of choline, PC and GPC. These results suggest that the H_2_O_2_ treatment disrupts the integrity of lipid membranes in C2C12 myoblasts without influencing the levels of choline and GPC, and TMAO treatment substantially improved the integrity of the lipid membranes in the cells through inhibiting the decomposition of PPC and reducing the level of lipid peroxidation.

Finally, we only analyzed the effect of TMAO in alleviating oxidative stress damage in myoblasts. It is unclear whether TMAO can rescue H_2_O_2_-impaired metabolic pathways and reduce oxidative stress damage in myotubes and skeletal muscle. This issue will be addressed in our future work. In addition, only the aqueous metabolites extracted from C2C12 myoblasts were analyzed herein. The combined analysis of hydrophobic and aqueous metabolites would provide a more global metabolic profile. Usually, hydrophobic metabolites are often associated with poor NMR spectra, which will make it difficult to calculate the integrals of most hydrophobic metabolites accurately. In the future, we would make efforts to optimize the NMR spectra of hydrophobic metabolites.

## 5. Conclusions

We demonstrated the beneficial effects and metabolic response of TMAO treatment on H_2_O_2_-impaired C2C12 myoblasts for the first time. TMAO treatment partly recovers the decreased proliferation and differentiation potential of the H_2_O_2_-impaired C2C12 cells, and significantly enhances the antioxidant activities in the H_2_O_2_-impaired cells by partly scavenging oxidative stress products. Furthermore, the H_2_O_2_ treatment remarkably decreased the levels of BCAAs and several amino acids including alanine, glycine, threonine, phenylalanine and histidine, and distinctly increased the level of PC related to the integrity of lipid membranes in C2C12 myoblasts, while the TMAO treatment significantly reversed the changing trends of these metabolite levels, thereby maintaining the stability of the cell membranes. This study is beneficial to the exploration of TMAO treatment as a promising strategy to alleviate oxidative stress damage in skeletal muscle.

## Figures and Tables

**Figure 1 biomolecules-12-01288-f001:**
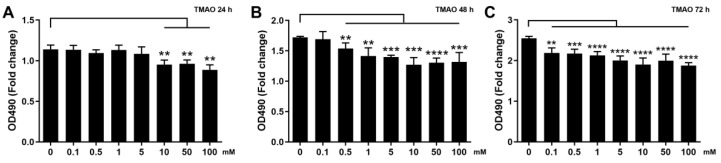
Effects of TMAO treatment on proliferation abilities of C2C12 myoblasts. (**A**) 24 h TMAO treatment. (**B**) 48 h TMAO treatment. and (**C**) 72 h TMAO treatment. Cell viabilities were used to represent the proliferation abilities of cells and assessed by MTS based on measured OD_490nm_ values (*n* = 6). Statistical significances: ** *p* < 0.01, *** *p* < 0.001, **** *p* < 0.0001.

**Figure 2 biomolecules-12-01288-f002:**
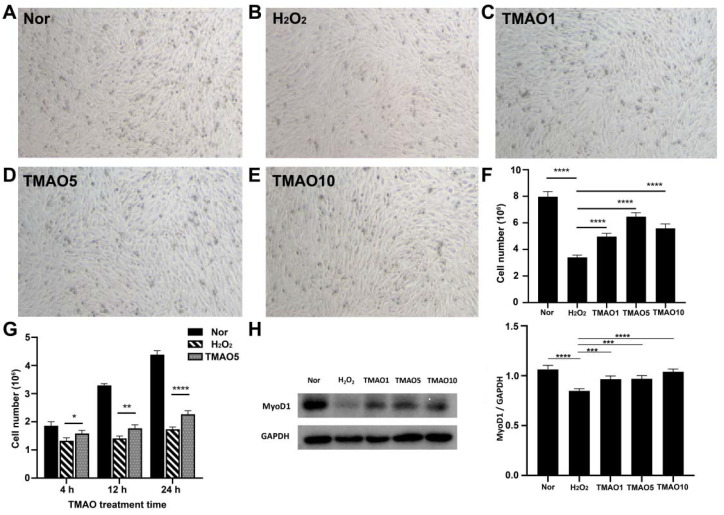
TMAO treatment enhanced proliferation abilities of H_2_O_2_-impaired C2C12 myoblasts. (**A**–**E**) Cell morphology of normal cells, H_2_O_2_-impaired cells and H_2_O_2_-impaired cells with 24 h TMAO treatment. (**F**) Cell numbers of the five groups of cells analyzed by cell counting (*n* = 3). (**G**) Cell number changed with 5 mM TMAO treatment within 24 h analyzed by cell counting (*n* = 3). (**H**) Expression levels of MyoD1 in the cells analyzed by Western blot. Statistical significances: * *p* < 0.05, ** *p* < 0.01, *** *p* < 0.001, **** *p* < 0.0001.

**Figure 3 biomolecules-12-01288-f003:**
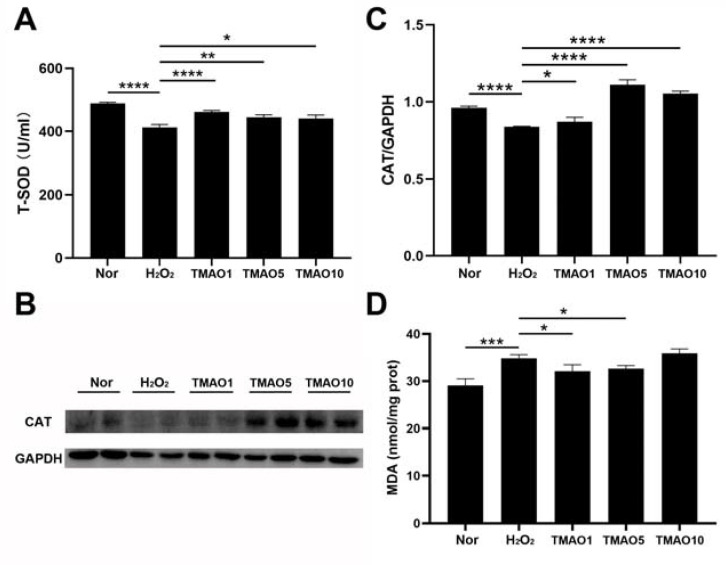
TMAO promotes the H_2_O_2_-induced decrease in antioxidant enzyme activity in C2C12 myoblasts. (**A**) Activities of T-SOD; (**B**) Western blot of CAT; (**C**) densitometry analysis of CAT expression level; and (**D**) levels of MDA. Statistical significances: * *p* < 0.05, ** *p* < 0.01, *** *p* < 0.001, **** *p* < 0.0001.

**Figure 4 biomolecules-12-01288-f004:**
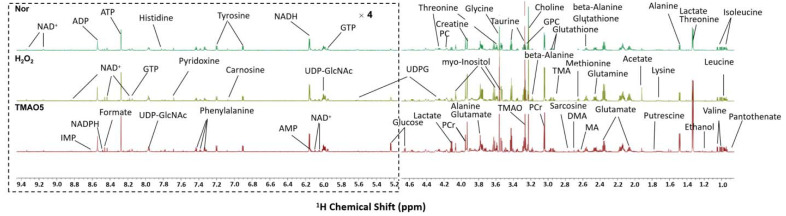
Typical 850 MHz ^1^H NMR spectra recorded on aqueous extracts derived from the three groups of C2C12 myoblasts.

**Figure 5 biomolecules-12-01288-f005:**
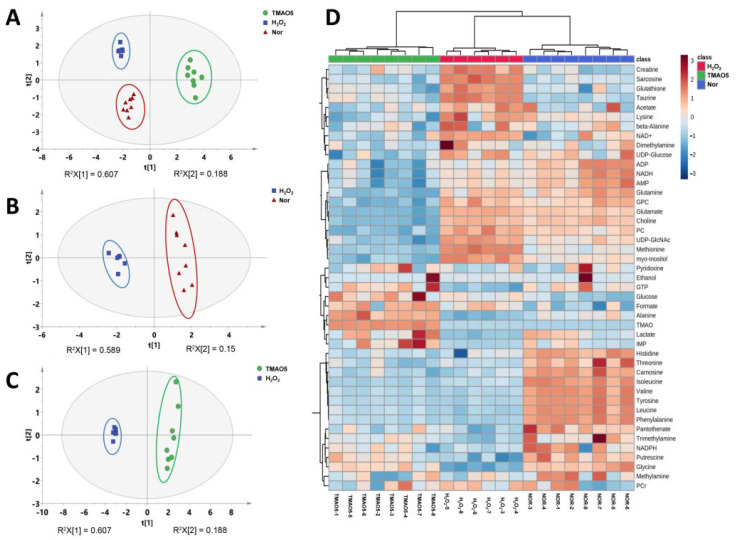
Multivariate analysis of ^1^H NMR spectra recorded on aqueous extracts derived from the three groups of C2C12 myoblasts. (**A**) PCA scores plot of the three groups of cells. (**B**) PCA scores plot of the H_2_O_2_ group vs. the normal group. (**C**) PCA scores plot of the TMAO5 group vs. the H_2_O_2_ group. These ellipses indicate the 95% confidence limits. (**D**) Hierarchical cluster analysis of the three groups of cells.

**Figure 6 biomolecules-12-01288-f006:**
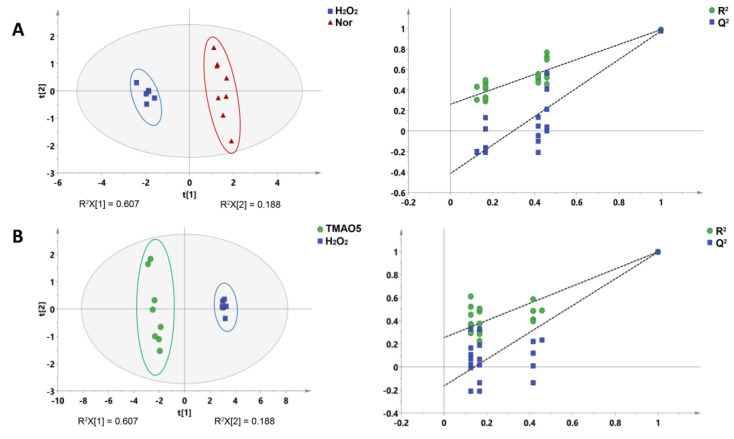
PLS-DA analysis for identifying significant metabolites primarily responsible for the discrimination of metabolic profiles among the three groups of C2C12 myoblasts. (**A**) PLS-DA scores plot of the H_2_O_2_ group vs. the normal group. (**B**) PLS-DA scores plot of the TMAO5 group vs. the H_2_O_2_ group. These ellipses indicate the 95% confidence limits.

**Figure 7 biomolecules-12-01288-f007:**
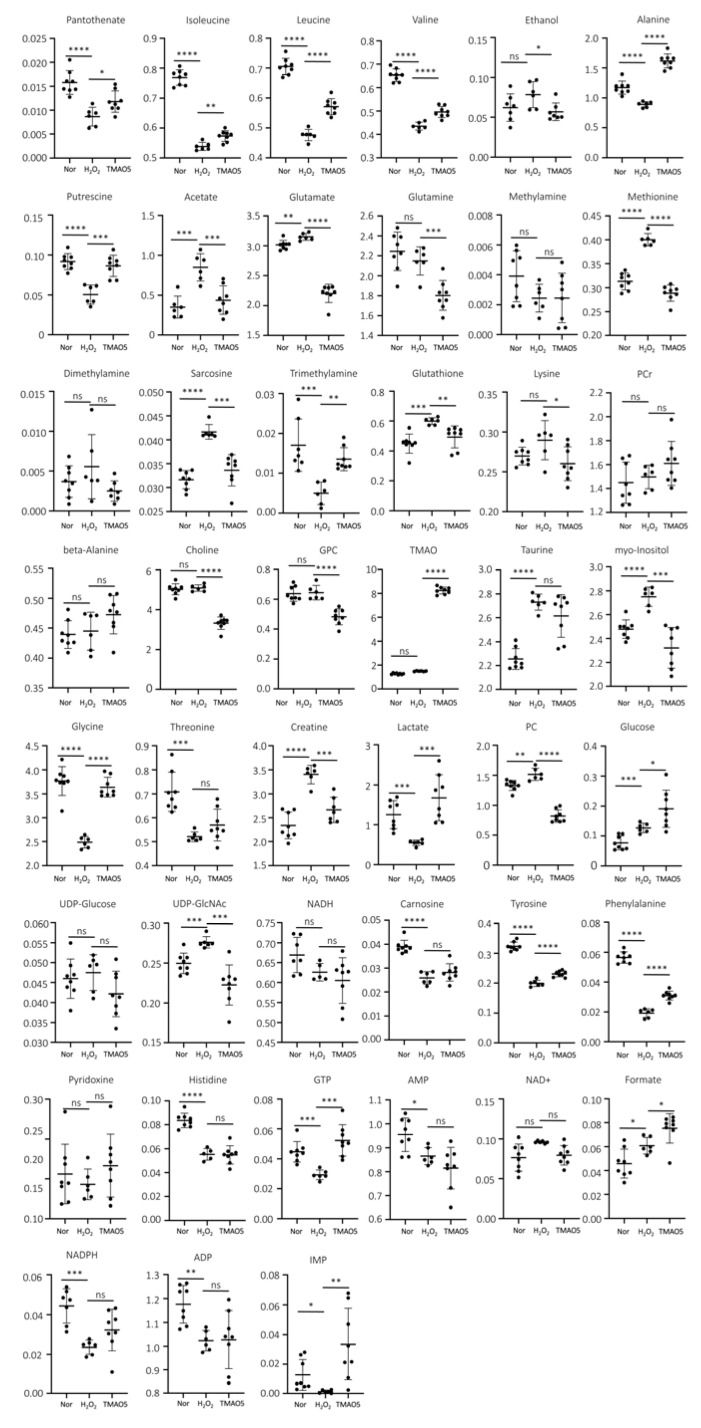
Multiple comparisons of metabolite levels between the three groups of C2C12 myoblasts based on relative NMR integrals. Statistical significances: ns, *p* > 0.05; *, *p* < 0.05; **, *p* < 0.01; ***, *p* < 0.001; ****, *p* < 0.0001.

**Figure 8 biomolecules-12-01288-f008:**
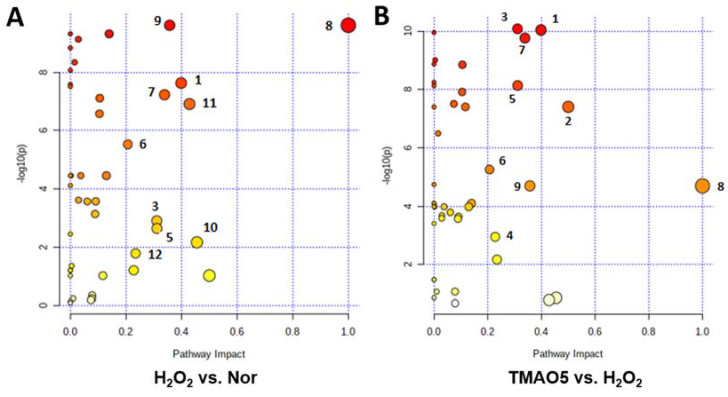
Significantly altered metabolic pathways identified from the pair-wise comparisons between the three groups of C2C12 myoblasts. (**A**) The H_2_O_2_ group vs. the normal group; (**B**) The TMAO5 group vs. the H_2_O_2_ group. The significantly altered metabolic pathways were identified with pathway impact values > 0.2 and *p* values < 0.05, which were labeled with the following numbers: (1) glutathione metabolism; (2) D-glutamine and D-glutamate metabolism; (3) alanine, aspartate and glutamate metabolism; (4) purine metabolism; (5) histidine metabolism; (6) arginine and proline metabolism; (7) glycine, serine and threonine metabolism; (8) phenylalanine, tyrosine and tryptophan biosynthesis; (9) phenylalanine metabolism; (10) beta-alanine metabolism; (11) taurine and hypotaurine metabolism; and (12) nicotinate and nicotinamide metabolism.

**Figure 9 biomolecules-12-01288-f009:**
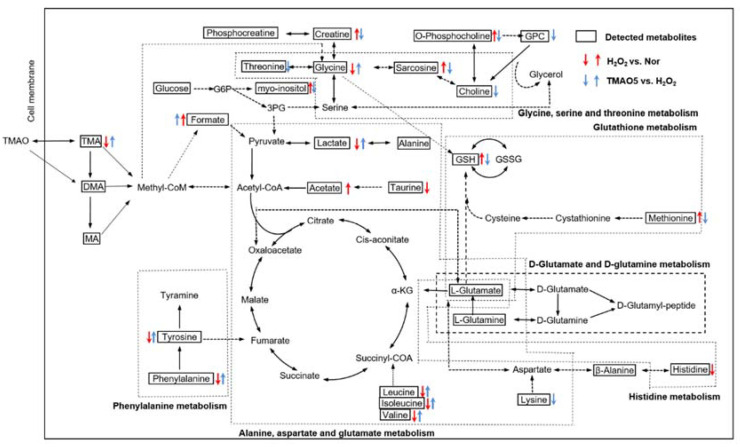
Schematic representation of significantly altered metabolic pathways identified from the pair-wise comparisons of H_2_O_2_ vs. normal, and TMAO5 vs. H_2_O_2_ based on the KEGG database. Dotted arrow indicate multiple biochemical reactions; solid arrows denote single biochemical reaction. The boxed metabolites and unboxed metabolites refer to detected and undetected metabolites, respectively. Abbreviations: TMAO, trimethylamine N-oxide; TMA, trimethylamine; DMA, dimethylamine; MA, methylamine; G6P, glucose-6-phosphate; 3PG, 3-phosphoglycerate; α-KG, α-ketoglutarate; GSH, reduced glutathione; GSSG; oxidized glutathione; and GPC, sn-Glycero-3-phosphocholine.

**Table 1 biomolecules-12-01288-t001:** Multiple comparisons of metabolite levels in the three groups of C2C12 cells based on relative NMR integrals.

Metabolites	Mean ± Standard			Multiple Comparisons		One-Way ANOVA
	Nor	H_2_O_2_	TMAO5	H_2_O_2_ vs. Nor	TMAO5 vs. H_2_O_2_	*p*
Pantothenate	0.016 ± 0.002	0.009 ± 0.002	0.012 ± 0.002	↓↓↓↓	↑	<0.0001
Isoleucine	0.769 ± 0.026	0.538 ± 0.013	0.573 ± 0.018	↓↓↓↓	↑↑	<0.0001
Leucine	0.705 ± 0.027	0.476 ± 0.019	0.571 ± 0.027	↓↓↓↓	↑↑↑↑	<0.0001
Valine	0.653 ± 0.027	0.435 ± 0.017	0.496 ± 0.024	↓↓↓↓	↑↑↑↑	<0.0001
Ethanol	0.062 ± 0.017	0.078 ± 0.016	0.057 ± 0.011	ns	↓	0.7696
Alanine	1.172 ± 0.110	0.890 ± 0.047	1.617 ± 0.118	↓↓↓↓	↑↑↑↑	<0.0001
Putrescine	0.092 ± 0.010	0.051 ± 0.012	0.086 ± 0.013	↓↓↓↓	↑↑↑	<0.0001
Acetate	0.552 ± 0.393	0.849 ± 0.172	0.437 ± 0.177	↑↑↑	↓↓↓	0.0370
Glutamate	3.015 ± 0.076	3.150 ± 0.058	2.203 ± 0.151	↑↑	↓↓↓↓	<0.0001
Glutamine	2.246 ± 0.193	2.148 ± 0.142	1.803 ± 0.148	ns	↓↓↓	<0.0001
Methylamine	0.004 ± 0.002	0.002 ± 0.001	0.002 ± 0.002	ns	ns	0.1214
Methionine	0.313 ± 0.017	0.400 ± 0.013	0.289 ± 0.017	↑↑↑↑	↓↓↓↓	<0.0001
DMA	0.004 ± 0.002	0.006 ± 0.004	0.002 ± 0.001	ns	ns	0.0747
Sarcosine	0.032 ± 0.002	0.042 ± 0.002	0.034 ± 0.003	↑↑↑↑	↓↓↓	<0.0001
TMA	0.020 ± 0.010	0.005 ± 0.003	0.014 ± 0.003	↓↓↓	↑↑	0.0022
Glutathione	0.450 ± 0.064	0.599 ± 0.023	0.494 ± 0.073	↑↑↑	↓↓	0.0007
Lysine	0.270 ± 0.011	0.290 ± 0.024	0.260 ± 0.021	ns	↓	0.0334
PCr	1.449 ± 0.172	1.497 ± 0.099	1.610 ± 0.185	ns	ns	0.1536
beta-Alanine	0.439 ± 0.023	0.445 ± 0.032	0.472 ± 0.032	ns	ns	0.0776
Choline	5.027 ± 0.275	5.092 ± 0.162	3.321 ± 0.314	ns	↓↓↓↓	<0.0001
GPC	0.637 ± 0.050	0.644 ± 0.049	0.482 ± 0.053	ns	↓↓↓↓	<0.0001
TMAO	1.272 ± 0.062	1.494 ± 0.027	8.242 ± 0.291	ns	↑↑↑↑	<0.0001
Taurine	2.255 ± 0.087	2.730 ± 0.068	2.615 ± 0.179	↑↑↑↑	ns	<0.0001
myo-Inositol	2.478 ± 0.078	2.750 ± 0.081	2.322 ± 0.170	↑↑↑↑	↓↓↓	<0.0001
Glycine	3.639 ± 0.205	2.487 ± 0.121	3.768 ± 0.299	↓↓↓↓	↑↑↑↑	<0.0001
Threonine	0.708 ± 0.082	0.520 ± 0.020	0.569 ± 0.067	↓↓↓	ns	<0.0001
Creatine	2.339 ± 0.280	3.404 ± 0.192	2.789 ± 0.428	↑↑↑↑	↓↓↓	<0.0001
Lactate	1.254 ± 0.353	0.546 ± 0.069	1.676 ± 0.578	↓↓↓	↑↑↑	0.0003
PC	1.332 ± 0.082	1.517 ± 0.102	0.821 ± 0.106	↑↑	↓↓↓↓	<0.0001
Glucose	0.077 ± 0.024	0.127 ± 0.016	0.191 ± 0.062	↑↑↑	↑	0.0001
UDPG	0.046 ± 0.005	0.047 ± 0.005	0.042 ± 0.006	ns	ns	0.1539
UDP-GlcNAc	0.250 ± 0.013	0.276 ± 0.007	0.222 ± 0.025	↑↑↑	↓↓↓	<0.0001
NADH	0.677 ± 0.046	0.619 ± 0.025	0.605 ± 0.057	ns	ns	0.0157
Carnosine	0.039 ± 0.003	0.026 ± 0.003	0.028 ± 0.004	↓↓↓↓	ns	<0.0001
Tyrosine	0.324 ± 0.014	0.199 ± 0.010	0.231 ± 0.010	↓↓↓↓	↑↑↑↑	<0.0001
Phenylalanine	0.241 ± 0.009	0.148 ± 0.007	0.177 ± 0.007	↓↓↓↓	↑↑↑↑	<0.0001
Pyridoxine	0.112 ± 0.075	0.086 ± 0.038	0.133 ± 0.079	ns	ns	0.4676
Histidine	0.084 ± 0.006	0.047 ± 0.021	0.055 ± 0.008	↓↓↓↓	ns	<0.0001
GTP	0.045 ± 0.007	0.029 ± 0.003	0.052 ± 0.010	↓↓↓	↑↑↑	0.0001
AMP	0.954 ± 0.070	0.865 ± 0.036	0.815 ± 0.087	↓	ns	0.0029
NAD+	0.076 ± 0.017	0.096 ± 0.002	0.079 ± 0.012	ns	ns	0.0251
Formate	0.046 ± 0.012	0.061 ± 0.007	0.075 ± 0.012	↑	↑	0.0002
NADPH	0.040 ± 0.015	0.024 ± 0.004	0.032 ± 0.010	↓↓↓	ns	0.0518
ADP	1.176 ± 0.079	1.023 ± 0.043	1.027 ± 0.123	↓↓	ns	0.0054
IMP	0.013 ± 0.011	0.001 ± 0.001	0.033 ± 0.024	↓	↑↑	0.0042

Statistical significances were determined by one-way ANOVA analysis followed by Tukey’s multiple comparison test. Statistical significances: ns, *p* > 0.05; ↓/↑, *p* < 0.05; ↓↓/↑↑, *p* < 0.01; ↓↓↓/↑↑↑, *p* < 0.001; ↓↓↓↓/↑↑↑↑, *p* < 0.0001. Abbreviations: DMA, dimethylamine; TMA, trimethylamine; PCr, phosphocreatine; GPC, sn-glycero-3-phosphocholine; TMAO, trimethylamine N-oxide; PC, O-phosphocholine; UDPG, UDP-glucose; UDP-GlcNAc, UDP-N-acetylglucosamine; GTP, guanosine triphosphate; AMP, adenosine monophosphate; NAD+, nicotinamide adenine dinucleotide; ADP, adenosine diphosphate; IMP, hypoxanthine nucleotide.

## Data Availability

Data are contained within the article or Appendix A.

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
