# Peer review of "NMR-Based Metabolomic Analysis for the Effects of Trimethylamine N-Oxide Treatment on C2C12 Myoblasts under Oxidative Stress"

_biomolecules, 2022, doi:10.3390/biom12091288_

Round 1

Reviewer 1 Report

The manuscript by Zou et al is well designed, pleasant to read with good scientific rigor. It will be a good fit in the Biomolecules journal. I have a few comments that the authors should address.

Specific comments:

1. Please refer to untreated cells as 'Control' or 'Normal' cells instead of Nor.

2. Please include information of statistical tests performed in the figure legends.

3. In figure 2, did the authors cultured the cells in differentiation media (DMEM +2% horse serum) before estimating MYOD? Or were the cells cultured in growth media (DMEM + 10% FBS)? For studying differentiation, cells are required to be cultured in growth media, and once they reach 85-90% confluency, the media has to be replaced with differentiation media. Also, MyHC and Myogenin are markers for differentiation and not MYOD. MYOD is expressed in activated satellite cells too. In fact, non-differentiating C2C12 cells  also express MYOD.

3. In figure 3, please rearrange the plots for better visualization. I feel it can be arranged as A, Activity of T-SOD B, Western blot of CAT, C, densitometry analysis of CAT expression level, and D, Levels of MDA.    

4. Line 327 has typo 'prolife'.

5. I feel the authors should bring Figure S6 in the main figure.

Reviewer 2 Report

In this manuscript, Zou and colleagues report on the metabolomic screening of trimethylamine N-oxide (TMAO) supplementation effects on proliferating C2C12 myoblasts. The nature of the intervention is a prospective “rescue” of the oxidative stress and loss of viability elicited by a 2-hour long H2O2 exposure. The study is elegant, well thought and interesting. The NMR analyses are remarkably well done and the NMR traces, as well as the MetaboAnalyst-derived analyses are quite compelling.

My only remark to the work relates to the setting of myoblasts in proliferation. The most important attribute of C2C12 myoblasts is their capacity of differentiating in myotubes. Considering that H2O2 and TMAO interventions are rapid (2hours and 24hours, respectively) and adaptable to C2C12 AFTER their differentiation in myotubes, I would like to ask the Authors to repeat their metabolomic analyses with H2O2 and TMAO AFTER a typical in vitro differentiation protocol. Generally, 7-10 days of serum starvation leads to a 70-90% fusion index in myotubes (this should be checked via immunostaining for myosin heavy chain). When C2C12 have formed homogenous cultures of myotubes, does TMAO rescue the H2O2-impaired metabolic pathways identified in the proliferative state to the same extent? Are the changes in amino acid and glutathione metabolism recapitulated in myotubes?

A minor point relates to Figures S6 and S7. I think Figure S6 should make its way to the main manuscript and be shown as main figure and not buried in the supplement. Figure S7 is not described in the Results, is only commented upon in the Discussion without experimental details. As such, the Figure is only confounding and inappropriately reported, and should therefore be removed.

Round 2

Reviewer 2 Report

The Authors did not address my question of even confirmatory analyses in differentiated myotubes. This reduces the overall levels of novelty, significance and merit of the paper. However, the paper in itself remains interesting and well done, and therefore I am recommending acceptance.